# Efficacy and safety of antimicrobial stewardship prospective audit and feedback in patients hospitalized with COVID-19: A protocol for a pragmatic clinical trial

Justin Z. Chen[1‡]*, Holly L. Hoang[1‡], Maryna Yaskina[2], Dima Kabbani[1], Karen E. Doucette[1], Stephanie W. Smith[1], Cecilia Lau[3], Jackson Stewart[3], Karen Zurek[4], Morgan Schultz[4], Carlos Cervera[1]

1 Division of Infectious Diseases, Department of Medicine, Faculty of Medicine & Dentistry, University of Alberta, Edmonton, Alberta, Canada, 2 Women and Children's Health Research Institute, University of Alberta, Edmonton, Alberta, Canada, 3 Pharmacy Services, Alberta Health Services, Edmonton, Alberta, Canada, 4 Pharmacy Services, Covenant Health, Edmonton, Alberta, Canada

‡ JZC and HLH are joint first authors on this work.
* jzchen@ualberta.ca

**Funding:** The author(s) received no specific funding for this work.

## Abstract

### Background

The use of broad-spectrum antibiotics is widespread in patients with COVID-19 despite a low prevalence of bacterial co-infection, raising concerns for the accelerated development of antimicrobial resistance. Antimicrobial stewardship (AMS) is vital but there are limited randomized clinical trial data supporting AMS interventions such as prospective audit and feedback (PAF). High quality data to demonstrate safety and efficacy of AMS PAF in hospitalized COVID-19 patients are needed.

### Methods and design

This is a prospective, multi-center, non-inferiority, pragmatic randomized clinical trial evaluating AMS PAF intervention plus standard of care (SOC) versus SOC alone. We include patients with microbiologically confirmed SARS-CoV-2 infection requiring hospital admission for severe COVID-19 pneumonia. Eligible ward beds and critical care unit beds will be randomized prior to study commencement at each participating site by computer-generated allocation sequence stratified by intensive care unit versus conventional ward in a 1:1 fashion. PAF intervention consists of real time review of antibacterial prescriptions and immediate written and verbal feedback to attending teams, performed by site-based AMS teams comprised of an AMS pharmacist and physician. The primary outcome is clinical status at post-admission day 15 measured using a 7-point ordinal scale. Patients will be followed for secondary outcomes out to 30 days. A total of 530 patients are needed to show a statistically significant non-inferiority, with 80% power and 2.5% one-sided alpha assuming standard deviation of 2 and the non-inferiority margin of 0.5.

**Competing interests:** The authors have declared that no competing interests exist.

## Discussion

This study protocol presents a pragmatic clinical trial design with small unit cluster randomization for AMS intervention in hospitalized COVID-19 that will provide high-level evidence and may be adopted in other clinical situations.

## Trial registration

This study is being performed at the University of Alberta and is registered at ClinicalTrials. gov (NCT04896866) on May 17, 2021.

## Introduction

COVID-19 is the disease caused by the severe acute respiratory coronavirus 2 (SARS-CoV-2), a novel coronavirus responsible for a global pandemic. The case burden and death toll is the highest of any respiratory virus outbreak in the modern antibiotic era [1]. Bacterial co-infections are known complications of viral pneumonia [2, 3] and it is estimated that 4–8% of hospitalized patients with COVID-19 will develop bacterial co-infection [4, 5]. The majority of COVID-19 management guidelines recommend judicious use of antimicrobials in patients presenting with pneumonia owing to the lack of benefit [6] and risks of *Clostridioides difficile* infection and other antimicrobial-associated adverse events [7, 8]. Despite this, significant and often broad-spectrum antibiotic use in hospitalized patients with COVID-19 is reported in the literature [9].

The COVID-19 pandemic is a significant source of unnecessary antibacterial therapy and is driving the often overlooked antimicrobial resistance (AMR) pandemic [10]. Many have highlighted the crucial role of formal antimicrobial stewardship program (ASP) involvement in managing the COVID-19 pandemic [11]. ASP goals are to combat AMR, reduce antimicrobial related complications, improve patient outcomes and maximize healthcare system efficiencies [5, 12, 13]. One core strategy is prospective audit and feedback (PAF) where antimicrobial stewardship (AMS) teams review patients' charts and provide real-time feedback to attending teams to optimize an antimicrobial prescription. This is a collaborative post-prescription strategy that course-corrects suboptimal prescribing. It serves as a clinical service that provides education and recommendations based on an individual patient's clinical context without providing direct clinical care.

While the benefit and safety of AMS PAF is described in settings such as community acquired pneumonia and viral acute respiratory infections, their cohort or quasi-experimental designs limit the ability to draw firm conclusions [14–18]. Given the significant antibiotic utilization in patients with COVID-19, this population offers a unique opportunity to study AMS PAF and produce high quality data using a robust randomized clinical trial design. There are no published studies to our knowledge that evaluate PAF in patients hospitalized with COVID-19 and specifically to determine the safety of rationalizing antibacterial therapy in those initiated empirically.

The objective of this study is to evaluate the safety and efficacy of PAF intervention plus standard of care (SOC) versus SOC alone in patients hospitalized with COVID-19 using clinical outcomes and a unique randomized pragmatic clinical trial design.

## Materials and methods

This is a prospective, multi-center, non-inferiority, pragmatic randomized clinical trial of PAF + SOC versus SOC alone in patients with microbiologically confirmed SARS-CoV-2 infection

| | STUDY PERIOD | | | | |
|---|---|---|---|---|---|
| | Enrollment | Allocation | Post-allocation | | Close-out |
| TIMEPOINT | *0* | *0* | *Post-admission day 15* | *30 days post-discharge* | *1 year* |
| **ENROLMENT:** | | | | | |
| *Eligibility screen* | X | | | | |
| *Allocation* | | X | | | |
| **INTERVENTIONS:** | | | | | |
| *Prospective audit and feedback + standard of care* | | | ●———————● | | |
| *Standard of care* | | | ●———————● | | |
| **ASSESSMENTS:** | | | | | |
| *Baseline variables:* | | | | | |
| *Attending unit* | | X | | | |
| *Attending specialty* | | X | | | |
| *Age* | | X | | | |
| *Sex* | | X | | | |
| *Weight* | | X | | | |
| *Charlson comorbidity index* | | X | | | |
| *SOFA and mSOFA score* | | X | | | |
| *Baseline ordinal scale* | | X | | | |
| *White blood cell count* | | X | | | |
| *Neutrophil count* | | X | | | |
| *Serum creatinine* | | X | | | |
| *C-reactive protein* | | X | | | |
| *Sputum culture* | | X | | | |
| *Respiratory pathogen panel* | | X | | | |
| *Blood cultures* | | X | | | |
| *Chest x-ray* | | X | | | |
| *Dexamethasone* | | X | | | |
| *Tocilizumab* | | X | | | |
| *Antibiotics* | | X | | | |
| *Primary outcome:* | | | | | |
| *7 point ordinal scale* | | | X | | |
| *Secondary outcomes:* | | | | | |
| *In-hospital mortality* | | | | X | |
| *30 day mortality* | | | | X | |
| *30 day readmission* | | | | X | |
| *Length of stay* | | | | X | |
| *Infectious disease consult* | | | | X | |
| *PAF recommendation type* | | | | X | |
| *Antibiotic indication* | | | | X | |
| *Antibiotic appropriateness* | | | | X | |
| *ASP recommendations* | | | | X | |
| *Number of PAF* | | | | X | |
| *PAF acceptance* | | | | X | |
| *Antibiotic utilization* | | | | X | |
| *30 day multi-drug resistant infection* | | | | X | |
| *30 day C. difficile infection* | | | | X | |
| *Neutropenia* | | | | X | |
| *Serum Creatinine* | | | | X | |

**Fig 1. SPIRIT schedule of enrollment, interventions, and assessments.**

in the preceding 2 weeks of hospitalization due to COVID-19. This study was reviewed and approved by the University of Alberta Research Ethics Board (Pro00105598) on January 27, 2021, and the Covenant Health Research Center on February 9, 2021. A waiver of individual patient consent was granted. This study is being performed at and sponsored by the University of Alberta and is registered at ClinicalTrials.gov (identifier, NCT04896866) on May 17, 2021 (protocol version 6.8, May 15, 2021). Study enrollment commenced in March 2021. We intend to publish the final results in a peer-reviewed medical journal. The study protocol adheres to the SPIRIT 2013 guidelines for clinical trials (Fig 1). Any modifications to the protocol will be reported to the University of Alberta Research Ethics Board and ClinicalTrials.gov, and will be stated in the final manuscript. We plan to grant public access to the full protocol, participant-level data, and the statistical code if required.

## Randomization

Each participating hospital will generate a line list of all hospital beds in adult COVID units and critical care units prior to enrollment. The line list will include the beds in each room on the unit and additional theoretical surge beds in the case of hospital surge and overcapacity during the pandemic. Randomization of beds will be stratified by COVID unit and critical care unit beds, and will be based on computer randomization with generation of allocation sequence in a 1:1 fashion to PAF + SOC or SOC alone.

If additional COVID or critical care units at a participating hospital are opened in the event of pandemic surge, the beds within the newly opened units will be included in the line list and the entire bed line list of the participating hospital will be subsequently re-randomized using the same randomization rules, to ensure appropriate stratification to COVID units and critical care units.

## Participating centers

Participating centers include:

1. University of Alberta Hospital, Edmonton, Alberta, Canada

2. Grey Nuns Community Hospital, Edmonton, Alberta, Canada

3. Misericordia Community Hospital, Edmonton, Alberta, Canada

At our study hospitals, physicians attend to patients geographically located on a single hospital ward rather than an assigned roster. Patients are frequently transferred between units for a variety of reasons such as changes in required level of care or infection control purposes. Furthermore, transient providers such as clinical associates, resident physicians, or physician extenders often provide overnight coverage. Patients are expected to have numerous physicians providing care through the course in hospital.

## Study population

The study target population is patients with SARS-CoV-2 infection requiring hospital admission for severe COVID-19 pneumonia. Patients with nosocomial-acquired SARS-CoV-2 infection were not included unless they were re-admitted to hospital from the community for severe COVID-19 pneumonia.

**Inclusion criteria.** All hospital beds in designated conventional units and critical care units accepting adult patients with microbiologically confirmed COVID-19 will be randomized. Patients are eligible for enrollment in the study if they meet all of the following inclusion criteria:

1. Age $\geq$ 18 years at the time of hospital admission.

2. Confirmed SARS-CoV-2 infection by nucleic acid testing or point-of-care antigen testing in the preceding 14 days of hospital admission.

3. Admitted from the community (including continuing care facilities).

4. Admitted to a hospital bed designated in the study.

5. A SpO2 $\leq$ 94% on room air, require supplemental oxygen, or have chest imaging findings compatible with COVID-19 pneumonia.

**Exclusion criteria.** All hospital beds outside of designated COVID units and critical care units will be excluded from randomization. A patient will be excluded from the study if:

1. The patient is enrolled in another clinical trial that involves antibacterial therapy.

2. The patient's goals of care are anticipated to be designated "total compassionate care" or palliative within 48 hours of admission.

3. The patient's progression to death is anticipated to be imminent and inevitable within 48 hours of admission.

4. The patient was attended by any member of the research team within 30 days of enrollment.

5. The patient is transferred from another acute care center.

**Participant recruitment.** Patients hospitalized with COVID-19 will be identified by the Alberta Health Services (AHS) Tableau dashboard, direct notification from site-based Infection Prevention & Control Programs, or direct screening of COVID units and critical care units. The AHS Tableau dashboard is a restricted access, secure dashboard developed by AHS analytics and reports positive COVID-19 cases that have been admitted to the study site along with date of admission and confirmatory SARS-CoV-2 test. The study team will screen patients every weekday (less statutory holiday) for study eligibility and enroll patients. The attending team and patients are blinded to the randomization sequence. Due to the nature of the PAF intervention, blinding of the ASP team will not be possible.

## Intervention

The intervention employed will be AMS PAF. PAF consists of an unsolicited review of active antimicrobial prescriptions with real time feedback to attending teams. Audits are performed prospectively, on weekdays less statutory holidays, by members of the ASP team consisting of infectious disease or AMS pharmacists and physicians. Verbal and written feedback will be provided in real time to attending team members (most often the attending physician) if the ASP team is making a specific recommendation. The attending physician is considered the most responsible physician and will make the final decision to accept or reject PAF recommendations, and is therefore minimal risk to patient safety. ASP teams do not conduct interviews or perform physical examinations with patients.

The initial PAF will occur on the day an eligible patient is identified and enrolled in the study. Follow-up audits will then occur weekly (+/-3 days to account for weekends or statutory holidays) and ad-hoc if a new antibacterial is prescribed, until the primary end-point at post-admission day 15. Appropriateness in antimicrobial prescribing will be assessed based on local clinical practice guidelines (AHS COVID-19 Scientific Advisory Group recommendations) [19]. If no such guidelines exist, then appropriateness is defined by expert opinion of the AMS team member(s) performing the audit. The focus of ASP recommendations will be to discontinue therapy where bacterial co-infection is not suspected or confirmed, and to optimize the duration and spectrum of antimicrobial therapy when antibiotics are warranted.

Only antibacterials will be audited and included in the analysis. Antimycobacterial, antiviral, antifungal, and antiparasitic agents will not be audited or included in the analysis. Prescriptions will be excluded from PAF if they are single doses or discontinued prior to PAF. Prescriptions will also be excluded from PAF and the final analysis if being used for surgical or medical prophylaxis.

Patients will be followed and analyzed in the arm they were assigned to regardless of transfers or movements through the hospitalization period. Patients will be followed out to 15 days where the primary endpoint will be assessed, and then up to 30 days for assessment of secondary endpoints.

## Primary outcome

The primary outcome will be the clinical status of the patient on post-admission day 15 measured using a 7-point ordinal scale, as presented in Table 1.

**Table 1. Primary outcome 7-point ordinal scale.**

| Clinical Outcome | Points |
|---|---|
| Not hospitalized, able to resume normal daily activities | 1 |
| Not hospitalized, unable to resume normal daily activities | 2 |
| Hospitalized, not on supplemental oxygen | 3 |
| Hospitalized, on supplemental oxygen | 4 |
| Hospitalized, on high flow oxygen therapy or non-invasive mechanical ventilation | 5 |
| Hospitalized, on ECMO or invasive mechanical ventilation | 6 |
| Death | 7 |

## Secondary outcomes

**Clinical endpoints.**   Hospital length of stay, in-hospital and 30-day mortality, *C. difficile* associated mortality, and 30-day re-admission rates will be examined.

**Antimicrobial stewardship endpoints.**   Antimicrobial utilization will be measured by days of therapy and length of therapy normalized by patient-days for the duration of hospitalization (capped at 30 days). Furthermore, the number of audits, types of recommendations, and rate of acceptance will be determined.

**Microbiologic endpoints.**   The 30-day multi-drug resistant (MDR) infection rates and 30-day *C. difficile* infection rate will be examined. The definition of MDR will be the lack of susceptibility to 1 or more agents in 3 or more antimicrobial categories active against the isolated bacteria [20]. In the case of *Staphylococcus aureus* and *Enterococcus* species, methicillin and vancomycin resistance, respectively, defines the strain as MDR regardless of resistance to other antimicrobials [20].

**Adverse events and complications.**   The 30-day rates of neutropenia and acute kidney injury, diagnosed and staged according to Kidney Disease Improving Global Outcomes definitions, will be examined.

## Data collection and management

The research team will assess, collect, and record all research data to a bespoke AHS REDcap® database in accordance with the protocol. All data access is controlled by unique usernames and passwords for individual study staff. Study staff will have access restricted to the functionality and data that are appropriate for their role in the study. Multiple education sessions were held to train the study staff regarding data integrity and quality. All study staff undergo mandatory AHS Information & Privacy education and training. Only investigators will have access to the final trial dataset. Research records will be kept for a minimum of 5 years in concordance with the University of Alberta Research Records Stewardship Guidance Procedure. This study does not involve the collection of biologic specimens.

## Sample size estimation

A total of 530 patients (265 per arm) are needed to show a statistically significant non-inferiority, with 80% power and 2.5% one-sided alpha assuming standard deviation of 2 and the non-inferiority margin of 0.5. The non-inferiority margin was estimated based on previous published data comparing tocilizumab plus SOC versus SOC alone, that included a primary endpoint based on the 7-level ordinal scale measured at day 15 [21]. In this study, the estimated mean score at day 15 in both groups was 3.04 and standard deviation 2.24. We opted for a conservative approach to meet the non-inferiority criteria selecting a non-inferiority margin less than 20%. A non-inferiority margin of 0.5 of the predicted mean score at day 15 (3.04) fulfilled

the requirements without leading to an unreasonably large sample size. Accounting for a 5% drop out rate, 279 participants will be recruited in each arm (558 participants total). When 250 participants from the control arm reach 15 days follow up or when 260 patients are recruited in the control arm (whichever is earlier) a non-comparative sample size reassessment will be performed. The standard deviation will be calculated for the control group and will be used to recalculate the sample size.

## Statistical analysis

All analyses will adhere to the principle of intention-to-treat (ITT). The ITT population will include all participants who were randomized in the trial. Additional analyses will be conducted on the per-protocol (PP) populations. The PP population will include all patients who completed the study as described in the protocol. This additional analysis will only be presented if there is a substantial difference in this populations compared to the ITT population.

The primary outcome will be a two-sample comparison of scores between the treatment and control arm. We will assess whether the scores in the treatment arm are not worse than in the control arm using the Mann-Whitney U test. A one-sided level of 0.025 will be used to declare significance for the non-inferiority. Results will be reported along with 95% confidence intervals.

Binary outcomes will be analyzed by a two-sample comparison of proportions using chi-square test. Continuous variables will be tested either by the Student's *t*-test or by the Wilcoxon rank sum test depending on whether assumptions for the *t*-test are satisfied. Fisher's exact test will be used to determine the statistical significance of difference with respect to the incidence of serious adverse events between the treatment and control arms. Baseline characteristics will be presented by the appropriate descriptive statistics: continuous variables will be summarized by mean, standard deviation, median, quartiles, minimum and maximum. Categorical data will be presented by absolute and relative frequencies (n and %).

All subgroup analyses will be considered exploratory. A time dependent and case intensity analyses will also be performed to determine if the ASP intervention has downstream effects on the prescribers. Comparison of the outcomes by sex (male/female), age group (by median age), and comorbidities will be performed. This analysis will be planned and described in statistical analysis plan. Primary and secondary outcomes will be adjusted for covariates. All adjusted analyses will be exploratory. Co-variates of interest will be included based on clinical relevance and will be specified in the statistical analysis plan. Adjustment will be performed by adding covariates to the original models. The senior biostatistician will be unblinded.

## Discussion

The Infectious Diseases Society of America (IDSA) guidelines for implementing an ASP recommends PAF as a core component of antimicrobial stewardship programs [22]. However, the evidence for AMS intervention, including PAF, is heterogeneous and generally of low quality. Conclusions are drawn from mostly cohort or quasi-experimental studies [16–18]. Furthermore, a systematic review of AMS interventional studies between 1950 and 2017 concluded that the quality of evidence has not improved with time and that limitations should inform the design of future stewardship studies [23].

The IDSA guidelines also implies PAF is performed on a daily basis. The guidelines recommend that if not feasible, limited PAF 3 times per week can still offer benefit. However many studies define PAF interventions as providing delayed feedback such as with performance report cards [24]. While this strategy is valid, heterogeneity amongst PAF studies make it

challenging to draw conclusions. Our study uses the daily PAF strategy as inferred by the IDSA guidelines as the intervention.

Antimicrobial stewardship programs identify various goals including to reduce rates of AMR, adverse events, and healthcare costs. Most published stewardship literature focuses on process outcomes rather than patient outcomes. Relatively few AMS studies report on mortality [18, 25, 26]. However, patient clinical outcomes including mortality should be emphasized more in stewardship research [27, 28]. Our study uses a 7-point ordinal scale of patient outcomes, including mortality, similar to other COVID-19 therapeutic trials as the primary outcome [29]. Furthermore, safety was prioritized as the primary outcome. In the setting of the first small-unit randomized clinical trial of prospective audit and feedback (PAF) in patients with COVID-19 to our knowledge, we believe it is critical to first demonstrate that rationalizing antibacterial therapy does not cause undue harm.

Few studies have examined the impact of AMS PAF in hospitalized patients using randomization. A Cochrane review of 221 studies identified only 58 RCTs by design [24]. Of only four studies with enablement plus feedback, only one intervened with real-time feedback. In another systematic review of 37 included stewardship interventional studies in hospital settings, only 3 of 14 studies evaluating PAF were RCT in design [30]. There are limitations to quasi-experimental and cohort study designs, the greatest is the limited ability to establish causal relationships due to multiple confounders that can influence antibiotic prescribing [23]. Robust randomization in AMS studies is desired but often not feasible resulting in studies using large unit cluster randomization such as by program, ward, or hospital. Traditional therapeutic trials intervene and study the outcomes in the same population with randomization at the individual participant level. In PAF studies, the intervention is on the prescriber whereas the outcomes of interest are in patients. At our 3 hospital study sites, the attending physician attends to a cohort of patients geographically located on a single ward. Therefore, randomizing by prescriber will effectively result in cluster randomization by ward. This may risk imbalanced arms as units have differences in patient acuity and medical complexity. Furthermore, differing stewardship practices in providers at baseline may also result in imbalanced arms. We therefore performed cluster randomization at the level of the hospital bed. To our knowledge, this is the lowest defined cluster reported in any AMS study that effectively minimizes contamination and baseline imbalances [26]. A robust homogeneous sample of patients by prescriber on every eligible COVID-19 patient is also achieved. More importantly, randomizing by hospital bed allows the same prescriber to potentially be in both arms of study which allows the evaluation of the effect of PAF independent of the prescriber.

High impact RCTs, especially studies using cluster randomization, have been performed without informed consent [31–36]. A waiver of consent was requested on the basis that our research involves no more than minimal risk to participants, alteration to consent requirements is unlikely to adversely affect the welfare of participants, and it is impossible or impracticable to carry out the research and to address the research question properly, given the research design, if the prior consent of participants is required. PAF functions as a reminder service to guide prescribers to follow patient-specific prescribing suggestions based on institutional guideline recommendations. PAF recommendations are reviewed and co-signed by the most responsible attending physician before the recommendations are live and executed. For this reason, PAF is considered no more than minimal risk. Furthermore, a waiver of consent minimizes selection bias of prescribers which is a threat to the validity of the research. A waiver of consent was granted by the University of Alberta Research Ethics Board pursuant to Article 3.7 of the Tri-Council Policy Statement: Ethical Conduct for Research Involving Humans– TCPS 2 (2018).

Given the open-label nature of our study, the Hawthorne effect is an unavoidable limitation. The Hawthorne effect is a phenomenon where prescribers may alter their antibiotic prescribing behavior once they are aware their prescribing is being monitored. The waiver of consent, a critical element of study design, may serve to minimize any Hawthorne effect. Additionally, physician learnings derived from the PAF + SOC arm may be applied to other patient prescriptions, including those patients in beds randomized to receive SOC. This is a reflection of the real world setting where the impact of PAF often extends beyond that individual encounter and begins to permeate a physician's regular prescribing and speaks to the pragmatic element of our study. We hypothesize this effect may be diluted by rapid staff turnover as many additional physicians have been brought in to manage the pandemic surge, providing care on a weekly rotational basis (often with a separate day and night team). Furthermore, there will be an emphasis on patient-specific feedback based on institutional COVID-19 management guidelines as opposed to generic recommendations. Only study team members will have knowledge of the randomization sequence such that physicians cannot anticipate which study arm they have been assigned.

Antimicrobial stewardship is fundamental in the COVID-19 pandemic response [11, 37]. However, there is emerging evidence that AMS resources are being diverted away and ASPs are impacted with reduced productivity [38]. This study aims, as a secondary objective, to demonstrate the value of AMS intervention in viral pandemic management.

## Conclusions

This study protocol describes a prospective, multi-centered, small unit cluster randomized, pragmatic clinical trial evaluating an antimicrobial stewardship intervention (prospective audit and feedback) in patients hospitalized with COVID-19 using a clinical primary outcome. The study design will provide high-level evidence and may be adopted in other clinical situations.

## Supporting information

**S1 Checklist. SPIRIT 2013 checklist: Recommended items to address in a clinical trial protocol and related documents.**
(DOC)

**S1 File. COVASP protocol approved by the University of Alberta Research Ethics Board (Pro00105598).**
(DOCX)

## Author Contributions

**Conceptualization:** Justin Z. Chen, Holly L. Hoang, Karen E. Doucette, Carlos Cervera.

**Data curation:** Maryna Yaskina.

**Formal analysis:** Maryna Yaskina.

**Investigation:** Justin Z. Chen, Holly L. Hoang, Dima Kabbani, Stephanie W. Smith, Cecilia Lau, Jackson Stewart, Karen Zurek, Morgan Schultz.

**Methodology:** Justin Z. Chen, Holly L. Hoang, Maryna Yaskina, Dima Kabbani, Karen E. Doucette, Carlos Cervera.

**Project administration:** Justin Z. Chen, Holly L. Hoang, Carlos Cervera.

**Supervision:** Karen E. Doucette, Carlos Cervera.

**Writing – original draft:** Justin Z. Chen, Holly L. Hoang.

**Writing – review & editing:** Justin Z. Chen, Holly L. Hoang, Maryna Yaskina, Dima Kabbani, Karen E. Doucette, Stephanie W. Smith, Cecilia Lau, Jackson Stewart, Karen Zurek, Morgan Schultz, Carlos Cervera.

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
