## [Decision Letter · Decision Letter 0]

11 Jan 2022

PONE-D-21-21326

Efficacy and Safety of antimicrobial stewardship prospective audit and feedback in patients hospitalized with COVID-19: a protocol for a pragmatic clinical trial

PLOS ONE

Dear Dr. Chen,

Thank you for submitting your manuscript to PLOS ONE. After careful consideration, we feel that it has merit but does not fully meet PLOS ONE’s publication criteria as it currently stands. Therefore, we invite you to submit a revised version of the manuscript that addresses the points raised during the review process.

The reviewers provided insougthful comments regarding the manuscript. Please provide answers for each of them.

We look forward to receiving your revised manuscript.

Kind regards,

Dafna Yahav

Academic Editor

PLOS ONE

Journal Requirements:

2. We have noted that the estimated number of participants is reported as 558 while in the protocol this is calculated at 530. Please could you clarify this discrepancy.

Reviewers' comments:

Reviewer's Responses to Questions

**Comments to the Author**

1. Does the manuscript provide a valid rationale for the proposed study, with clearly identified and justified research questions?

Reviewer #1: Yes

Reviewer #2: Partly

2. Is the protocol technically sound and planned in a manner that will lead to a meaningful outcome and allow testing the stated hypotheses?

Reviewer #1: Yes

Reviewer #2: Partly

3. Is the methodology feasible and described in sufficient detail to allow the work to be replicable?

Reviewer #1: Yes

Reviewer #2: Yes

4. Have the authors described where all data underlying the findings will be made available when the study is complete?

Reviewer #1: Yes

Reviewer #2: Yes

5. Is the manuscript presented in an intelligible fashion and written in standard English?

Reviewer #1: Yes

Reviewer #2: Yes

6. Review Comments to the Author

You may also provide optional suggestions and comments to authors that they might find helpful in planning their study.

Reviewer #1: Very nicely designed study and described in a clear manner, the authors address the challenges of such a study and examine two types of endopoints, both for the patient and the antibiotic stewardship outcomes.

my only major concern with this study is what the authors refer to as the Hawhtorne effect.

in my opinion it will be more severe than just changing of practice once one knows that he is being monitored.

the way the study is conducted the same physician can be randomized into the study in both arms at the same time on two different patients. ASP impacts clinical behavior and is never isolated to a single patient, clinicians learn and change their practice from case to case and i am sure that if a teaching point is successful in impacting a clinician to modify his behavior he will modify his behavior moving forward on other cases (if the ASP was successful) .

i fear that in the current study the investigators will not be able to assess this effect.

analysis should be performed not only per patient / bed but per clinician and in a time dependent manner to see if the ASP intervention has downstream effects on the prescriber.

Reviewer #2: The authors present a protocol for a pragmatic randomized controlled trial evaluating antimicrobial stewardship prospective audit and feedback in the context of COVID-19. Although there is growing evidence to support ASP PAF in general, more high quality data are needed, and COVID-19 is an optimal context to evaluate this important strategy. This is a much needed study but the authors should strengthen the argument for why it is needed, why they've selected a primary outcome that links more with safety (given that we already know not using antibiotics in the context of COVID-19 without co-infection is safe), and how they selected the non-inferiority margin. Additional suggestions below:

1. Abstract mentions confirmed SARS-CoV-2 prior to hospitalization, but will patients with nosocomial COVID-19 be included?

2. Please clarify in abstract - is randomization at the ward or patient bed level?

3. Abstract background can be shortened slightly in favour of additional detail on the intervention, e.g. who is providing feedback (interdisciplinary w/ pharmacist and physician?, how will feedback be provided?)

4. Abstract - what is the primary outcome and the non-inferiority margin?

5. Introduction - please cite examples of cohort and quasi experimental ASP studies in COVID-19.

6. More explanation is needed as to why this research is important. There are existing PAF/ASP studies in respiratory tract infections/CAP. Please explain why COVID-19 would be unique.

7. Lines 55-57. It seems that the authors have used secondary infection and co-infection interchangeably. Consider distinguishing the two in terms of their risk for bacterial infection.

8. "There is no interaction with the patient before, during, or after the intervention." I do not believe this is true for all PAF strategies, consider rephrasing.

9. Methods - are patients already being enrolled, at what date was the first patient enrolled?

10. How will "contamination" within providers be addressed? Presumably a provider could care for patients in SOC or SOC+PAF beds? A goal of PAF should be to empower providers to be stewards without the intervention from ASP/ID experts when the experts are not around. So if the intervention is done well, there should be a lot of within provider contamination over time. Cluster randomization at the provider level stratified by prescriber service would be ideal. This may need to be further addressed as a limitation to be mitigated.

11. Please provide more detail on the use of the primary outcome ordinal scale. Will it be based on change from baseline or simply the status of the patient at day 15?

12. Please define multi-drug resistant infection rates.

13. How was the non-inferiority margin of 0.5 selected?

14. Why was safety selected as a primary outcome? There doesn't seem to be any need to show that discontinuing antibiotics in COVID-19 is safe. There is already a Cochrane review on this topic. It may be more informative to make antibiotic utilization a primary outcome, to show that PAF is effective in the context of COVID-19. It is admirable that the authors select a clinical outcome as the primary outcome but ideally would want one that PAF can have a direct positive impact on (e.g., antibiotic-related harms, length of stay).

7. PLOS authors have the option to publish the peer review history of their article (what does this mean?). If published, this will include your full peer review and any attached files.

Reviewer #1: No

Reviewer #2: No

---

## [Author Response · Author response to Decision Letter 0]

15 Feb 2022

Dear Dr. Yahav,

RE: PLOS ONE Decision: Revision required [PONE-D-21-21326] - [EMID:77fa8cea8ed2289f]

We thank PLOS ONE and the reviewers for reviewing this manuscript, providing thoughtful feedback and allowing the opportunity to submit revisions to improve the quality of the product. We have reviewed the reviewers’ comments carefully and have revised the manuscript to answer the reviewers’ questions and to incorporate suggestions. Please find our responses below. We have also resubmitted the revised manuscript with and without track changes as requested. Please note references to lines are to the revised, unmarked version of the manuscript without track changes.

Journal Requirements:

We have carefully reviewed the style requirements. Revisions include the formatting of the title, authors, affiliations, figure titles, headings, in-text citations, and references. The rebuttal letter, marked, and unmarked revised manuscript files are named according to journal requirements. 

2. We have noted that the estimated number of participants is reported as 558 while in the protocol this is calculated at 530. Please could you clarify this discrepancy.

A total of 530 participants (265 per arm) with the completed data for the final analysis are needed to show a statistically significant non-inferiority, with 80% power and 2.5% one-sided alpha assuming standard deviation of 2 and the non-inferiority margin of 0.5. We increased the sample size by 5% (558 participants) to account for the risk of participant exclusions or missing data.

The protocol manuscript does not contain data and the data availability policy is not applicable to our article.

Reviewer #1: Very nicely designed study and described in a clear manner, the authors address the challenges of such a study and examine two types of endopoints, both for the patient and the antibiotic stewardship outcomes.

my only major concern with this study is what the authors refer to as the Hawhtorne effect.

in my opinion it will be more severe than just changing of practice once one knows that he is being monitored.

the way the study is conducted the same physician can be randomized into the study in both arms at the same time on two different patients. ASP impacts clinical behavior and is never isolated to a single patient, clinicians learn and change their practice from case to case and i am sure that if a teaching point is successful in impacting a clinician to modify his behavior he will modify his behavior moving forward on other cases (if the ASP was successful) .

i fear that in the current study the investigators will not be able to assess this effect.

analysis should be performed not only per patient / bed but per clinician and in a time dependent manner to see if the ASP intervention has downstream effects on the prescriber.

Thank you for your feedback. Your point regarding learned behaviors contaminating the standard-of-care arm is well taken and has been acknowledged in the Discussion (lines 324-333). We believe this effect is likely diluted based on how our study 3 hospitals function. At our study hospitals, physicians attend to patients geographically located on a single hospital ward rather than an assigned roster. Patients are frequently transferred between units for a variety of reasons such as changes in required level of care or infection control purposes. Furthermore, transient providers such as clinical associates, resident physicians, or physician extenders often provide overnight coverage. For these reasons, patients are expected to have numerous physicians providing care through the course in hospital. This has been clarified in Methods (lines 114-118). Any behavior modifications derived from ASP effect in one attending physician is unlikely to impact the next attending physician. 

We will perform time dependent and case intensity analyses to see if the ASP intervention has downstream effects on the prescribers was added to the Methods (lines 254-255). Antimicrobial utilization (secondary outcome) analysis will also provide insight.

Reviewer #2: The authors present a protocol for a pragmatic randomized controlled trial evaluating antimicrobial stewardship prospective audit and feedback in the context of COVID-19. Although there is growing evidence to support ASP PAF in general, more high quality data are needed, and COVID-19 is an optimal context to evaluate this important strategy. This is a much needed study but the authors should strengthen the argument for why it is needed, why they've selected a primary outcome that links more with safety (given that we already know not using antibiotics in the context of COVID-19 without co-infection is safe), and how they selected the non-inferiority margin. Additional suggestions below:

Thank you for your review and thoughtful feedback. We believe the specific concerns are addressed below in suggestion #6, 14, and 13 respectively.

1. Abstract mentions confirmed SARS-CoV-2 prior to hospitalization, but will patients with nosocomial COVID-19 be included?

We include patients with microbiologically confirmed SARS-CoV-2 infection requiring hospital admission for severe COVID-19 pneumonia. Patients with nosocomial-acquired SARS-CoV-2 infection were not included unless they were re-admitted to hospital from the community for severe COVID-19 pneumonia. We have added this clarification in the Methods (lines 121-123). 

2. Please clarify in abstract - is randomization at the ward or patient bed level?

Randomization is at the patient bed level. We have changed “Eligible ward and critical care unit beds will be randomized…” to “Eligible ward beds and critical care unit beds will be randomized…” to provide clarity (lines 29-31).

3. Abstract background can be shortened slightly in favour of additional detail on the intervention, e.g. who is providing feedback (interdisciplinary w/ pharmacist and physician?, how will feedback be provided?)

The Abstract background has been shortened in favor of adding the following additional detail: “PAF intervention consists of real time review of antibacterial prescriptions and immediate written and verbal feedback to attending teams, performed by site-based AMS teams comprised of an AMS pharmacist and physician” (lines 31-33).

4. Abstract - what is the primary outcome and the non-inferiority margin?

The Abstract has been revised to incorporate the primary outcome (lines 33-34) and non-inferiority margin (line 37). 

5. Introduction - please cite examples of cohort and quasi experimental ASP studies in COVID-19.

The Introduction has been revised to include 2 citations (citations 14 and 15) referencing ASP studies using cohort and quasi-experimental designs (line 70). To our knowledge, there is no published primary literature describing ASP prospective audit and feedback of antibacterials in patients with COVID-19.

6. More explanation is needed as to why this research is important. There are existing PAF/ASP studies in respiratory tract infections/CAP. Please explain why COVID-19 would be unique.

While the benefit and safety of ASP/PAF studies in community acquired pneumonia and viral acute respiratory infections has been studied, their cohort or quasi-experimental designs limit the ability to draw firm conclusions. Currently, the COVID-19 pandemic is a significant source of unnecessary antibacterial therapy and is driving the antimicrobial resistance pandemic. Antimicrobial stewardship intervention, supported by high quality clinical trial data, is warranted to control the pandemic within a pandemic. The Introduction has been extensively revised to incorporate detail and explanation (lines 47-78):

COVID-19 is the disease caused by the severe acute respiratory coronavirus 2 (SARS-CoV-2), a novel coronavirus responsible for a global pandemic. The case burden and death toll is the highest of any respiratory virus outbreak in the modern antibiotic era. Bacterial co-infections are known complications of viral pneumonia. It is estimated that 4-8% of hospitalized patients with COVID-19 will develop bacterial co-infection. The majority of COVID-19 management guidelines recommend judicious use of antimicrobials in patients presenting with pneumonia owing to the lack of benefit and risks of Clostridioides difficile infection and other antimicrobial-associated adverse events. Despite this, significant and often broad-spectrum antibiotic use in hospitalized patients with COVID-19 is reported in the literature.

The COVID-19 pandemic is a significant source of unnecessary antibacterial therapy and is driving the often overlooked antimicrobial resistance (AMR) pandemic. Many have highlighted the crucial role of formal antimicrobial stewardship program (ASP) involvement in managing the COVID-19 pandemic. ASP goals are to combat AMR, reduce antimicrobial related complications, improve patient outcomes and maximize healthcare system efficiencies. One core strategy is prospective audit and feedback (PAF) where antimicrobial stewardship (AMS) teams review patients' charts and provide real-time feedback to attending teams to optimize an antimicrobial prescription. This is a collaborative post-prescription strategy that course-corrects suboptimal prescribing. It serves as a clinical service that provides education and recommendations based on an individual patient’s clinical context without providing direct clinical care. 

While the benefit and safety of AMS PAF is described in settings such as community acquired pneumonia and viral acute respiratory infections, their cohort or quasi-experimental designs limit the ability to draw firm conclusions. Given the significant antibiotic utilization in patients with COVID-19, this population is an opportunity to the study of AMS PAF to produce high quality data using a robust randomized clinical trial design. There are no published studies to our knowledge that evaluate PAF in patients hospitalized with COVID-19 and specifically to determine the safety of rationalizing antibacterial therapy in those initiated empirically. 

The objective of this study is to evaluate the safety and efficacy of PAF intervention plus standard of care (SOC) versus SOC alone in patients hospitalized with COVID-19 using clinical outcomes and a unique randomized pragmatic clinical trial design.

7. Lines 55-57. It seems that the authors have used secondary infection and co-infection interchangeably. Consider distinguishing the two in terms of their risk for bacterial infection.

The use of “secondary bacterial infection” has been revised to “bacterial co-infection” in the manuscript (line 49 and line 172).

8. "There is no interaction with the patient before, during, or after the intervention." I do not believe this is true for all PAF strategies, consider rephrasing.

Thank you for pointing this out. We removed the line "There is no interaction with the patient before, during, or after the intervention" from the Introduction.

At our 3 study sites, the antimicrobial stewardship teams perform chart reviews and provide written and verbal feedback to the attending team. The antimicrobial stewardship team do not conduct an interview or physical examination with the patient. In the Methods, we revised the text “There is no interaction with the patient before, during, or after the intervention” to “ASP teams do not conduct interviews or perform physical examinations with patients” (line 164).

9. Methods - are patients already being enrolled, at what date was the first patient enrolled?

We have added “Study enrollment commenced in March 2021.” to the Methods (lines 87-88).

10. How will "contamination" within providers be addressed? Presumably a provider could care for patients in SOC or SOC+PAF beds? A goal of PAF should be to empower providers to be stewards without the intervention from ASP/ID experts when the experts are not around. So if the intervention is done well, there should be a lot of within provider contamination over time. Cluster randomization at the provider level stratified by prescriber service would be ideal. This may need to be further addressed as a limitation to be mitigated.

We had considered cluster randomization at the provider level during study conception and design. However, at all 3 of our study hospitals, one physician attends a geographic a unit (average 20 beds per unit) for 1 week at a time. By randomizing at the provider level, cluster randomization at the unit level would have inadvertently occurred which may risk imbalanced arms as units have differences in acuity and medical complexity. Furthermore, differing stewardship practices in providers at baseline may also result in imbalanced arms. We therefore performed cluster randomization at the level of the hospital bed. To our knowledge, this is the lowest defined cluster reported in any AMS study that effectively minimizes contamination and baseline imbalances. This has been clarified in the Discussion (lines 297-303).

Your point regarding learned behaviors contaminating the standard-of-care arm is well taken and has been acknowledged in the Discussion (lines 324-333). We believe this effect is likely diluted based on how our study 3 hospitals function. At our study hospitals, physicians attend to patients geographically located on a single hospital ward rather than an assigned roster. Patients are frequently transferred between units for a variety of reasons such as changes in required level of care or infection control purposes. Furthermore, transient providers such as clinical associates, resident physicians, or physician extenders often provide overnight coverage. For these reasons, patients are expected to have numerous physicians providing care through the course in hospital. This has been clarified in Methods (lines 114-118). Any behavior modifications derived from ASP effect in one attending physician is unlikely to impact the next attending physician. 

We will perform time dependent and case intensity analyses to see if the ASP intervention has downstream effects on the prescribers was added to the Methods (lines 254-255). Antimicrobial utilization (secondary outcome) analysis will also provide insight.

11. Please provide more detail on the use of the primary outcome ordinal scale. Will it be based on change from baseline or simply the status of the patient at day 15?

The primary outcome will be the clinical status of the patient on post-admission day 15 measured using a 7-point ordinal scale. The Methods has been revised to provide clarity (lines 186-187). 

12. Please define multi-drug resistant infection rates.

We will use the definition of multi-drug resistance (MDR) as the lack of susceptibility to one or more agents in three or more antimicrobial categories active against the isolated bacteria [1]. In the case of Staphylococcus aureus and Enterococcus species, methicillin and vancomycin resistance, respectively, defines the strain as MDR regardless of resistance to other antimicrobials [1].

This has been added to the Methods (lines 201-204).

13. How was the non-inferiority margin of 0.5 selected?

The following has been added to the Methods (lines 222-227): “The non-inferiority margin was estimated based on previous published data comparing tocilizumab plus standard of care versus standard of care alone, that included a primary end-point based on the 7-level ordinal scale measured at day 15 [2]. In this study, the estimated mean score at day 15 in both groups was 3.04 and standard deviation 2.24. We opted for a conservative approach to meet the non-inferiority criteria selecting a non-inferiority margin less than 20%. A non-inferiority margin of 0.5 of the predicted score at day 15 (3.04) fulfilled the requirements without leading to an unreasonably large sample size.”

14. Why was safety selected as a primary outcome? There doesn't seem to be any need to show that discontinuing antibiotics in COVID-19 is safe. There is already a Cochrane review on this topic. It may be more informative to make antibiotic utilization a primary outcome, to show that PAF is effective in the context of COVID-19. It is admirable that the authors select a clinical outcome as the primary outcome but ideally would want one that PAF can have a direct positive impact on (e.g., antibiotic-related harms, length of stay).

There is a body of evidence demonstrating the incidence of bacterial co-infection in patients hospitalized with COVID-19 is low, suggesting routine empiric antibacterial therapy targeting bacterial coinfection is likely not necessary unless there is strong suspicion or evidence that one is present. The Cochrane review includes 11 randomized clinical studies of over 11 thousand patients investigating antibiotics compared to placebo, standard of care alone, or another antibiotic for treatment of COVID-19 [3]. Azithromycin was the only antimicrobial studied. This Cochrane review concludes that 28 day mortality is not reduced with azithromycin treatment. It however does not examine discontinuation of antibacterial therapy initiated by a prescriber.

Empiric, broad-spectrum antibiotics remain commonly prescribed in patients with COVID-19 upon hospitalization. A number of these patients are likely to go on to “complete a course” of antibiotics despite not exhibiting any features of bacterial co-infection. Clinical practice often does not follow guideline recommendations. There are no high quality evidence to draw firm conclusions that antibacterial therapy can be safely discontinued in those initiated empirically.

In this context, and in the setting of the first small unit randomized clinical trial of prospective audit and feedback (PAF) in patients with COVID-19 to our knowledge, we believe it is critical to first demonstrate that rationalizing antibacterial therapy does not cause undue harm in patients. We have added this to the Discussion (lines 282-285).

Other outcomes that PAF can have a direct positive impact on, such as antibiotic utilization, antibiotic-related harms, and length of stay, remain important and are examined as secondary outcomes.

On behalf of all the authors, we would again like to thank PLOS ONE and the reviewers for taking the time to review our manuscript.

---

## [Editor Report · Decision Letter 1]

3 Mar 2022

Efficacy and Safety of antimicrobial stewardship prospective audit and feedback in patients hospitalized with COVID-19: a protocol for a pragmatic clinical trial

PONE-D-21-21326R1

Dear Dr. Chen,

We’re pleased to inform you that your manuscript has been judged scientifically suitable for publication and will be formally accepted for publication once it meets all outstanding technical requirements.

Kind regards,

Dafna Yahav

Academic Editor

PLOS ONE
---

## [Editor Report · Acceptance letter]

14 Mar 2022

PONE-D-21-21326R1 

Efficacy and Safety of antimicrobial stewardship prospective audit and feedback in patients hospitalized with COVID-19: a protocol for a pragmatic clinical trial 

Dear Dr. Chen:

I'm pleased to inform you that your manuscript has been deemed suitable for publication in PLOS ONE. Congratulations! Your manuscript is now with our production department. 

Kind regards, 

on behalf of

Dr. Dafna Yahav 

Academic Editor

PLOS ONE